# The Effectiveness and Usability of a Rehabilitation Program Using the Nintendo Switch to Promote Healthy Aging in Older People with Cognitive Impairment: A Randomized Clinical Trial

**DOI:** 10.3390/healthcare12060672

**Published:** 2024-03-16

**Authors:** Melissa Helen Zegarra-Ramos, Cristina García-Bravo, Elisabet Huertas-Hoyas, Gemma Fernández-Gómez, Mª Pilar Rodríguez-Pérez, Jorge Pérez-Corrales, Sara García-Bravo

**Affiliations:** 1Orpea Collado Villalba, Nursing Home, 28400 Madrid, Spain; mh.zegarra.2017@alumnos.urjc.es; 2Department of Physical Therapy, Occupational Therapy, Physical Medicine and Rehabilitation, Universidad Rey Juan Carlos, 28922 Alcorcón, Spain; cristina.bravo@urjc.es (C.G.-B.); gemma.fernandez@urjc.es (G.F.-G.); pilar.rodriguez@urjc.es (M.P.R.-P.); jorge.perez@urjc.es (J.P.-C.); sara.garcia.bravo@urjc.es (S.G.-B.); 3Research Group in Evaluation and Assessment of Capacity, Functionality and Disability (TO + IDI), Universidad Rey Juan Carlos, 28922 Alcorcón, Spain; 4Research Group of Humanities and Qualitative Research in Health Science (Hum&QRinHS), Universidad Rey Juan Carlos, 28922 Alcorcón, Spain; 5Physiocare Madrid, Physiotherapy Clinic, 28026 Madrid, Spain

**Keywords:** occupational therapy, Nintendo Switch, healthy aging, quality of life, elderly people

## Abstract

(1) Background: Healthy aging is the process of developing and maintaining functional capacity and optimizing involvement in order to improve one’s quality of life as people age. This study aimed to encourage healthy aging in people with cognitive impairment, as well as a control group, via the use of the Nintendo Switch combined with occupational therapy sessions, aiming to improve gross and fine motor skills, functionality, quality of life, and cognitive status. (2) Methods: A randomized clinical trial was undertaken. The sample was randomized using the OxMar software program Attribution 4.0 International, facilitating the division into a control group (CG), who received conventional occupational therapy sessions, and an experimental group (EG), who received therapy incorporating the Nintendo Switch, in addition to conventional occupational therapy sessions. The intervention period with the Nintendo Switch lasted for 8 weeks. (3) Results: Thirty-two participants were included in the study. Significant differences were found in the vast majority of the variables analyzed, which shows an improvement following the intervention; these improvements were mainly observed in measures of skill and the perception of quality of life. (4) Conclusions: An eight-week intervention with the Nintendo Switch alongside conventional occupational therapy helps to maintain cognitive status and functional independence. Following 8 weeks of intervention with the Nintendo Switch, improvements in gross motor dexterity, fine motor dexterity, and quality of life were observed in older people with cognitive impairment.

## 1. Introduction

Normal aging is defined as a heterogeneous and modifiable process that is associated with a complex pattern of gains and losses in cognitive and affective domains [1,2]. Aging is the most significant risk factor in most common chronic diseases [3]. Some cognitive abilities, such as working memory and processing speed, tend to decline with age [1]. However, it is possible for a brain to age and be healthy simultaneously [4].

According to the World Health Organization, healthy aging is more than just the absence of disease. It is “the process of developing and maintaining functional capacity that enables well-being in old age” [5]. Healthy aging involves the ability to maintain vitality, independence, and quality of life, regardless of diseases or social characteristics that can be associated with health status [6]. The rehabilitation process can help older people to improve their habits and enhance healthy aging [6,7]. Occupational therapy focuses on finding out the person’s interests and identifying the environmental supports and occupational modifications needed to enhance healthy aging [7].

Digital technologies, used in rehabilitation and health promotion processes, have implications for functioning in daily life, thus prolonging independent living in older people. However, age-related deterioration in motor and cognitive abilities can lead to the emergence of barriers, hindering the independent use of new technologies [8]. The implementation of new technologies via virtual reality (VR) may be an appropriate way to promote healthy aging in older people [9].

It has been shown that VR-based video games have physical benefits via improving general mobility or hand–eye coordination, cognitive benefits like attention or memory, and social benefits through the promotion of social relationships and spontaneity [10,11,12]. In older people, VR benefits have also been observed with the use of the Kinect [13,14] or the Wii [15,16].

Currently, there are no studies that implement conventional occupational therapy treatment combined with the use of VR through the Nintendo Switch in older people to promote healthy aging. However, it has been shown that, in other pathologies such as multiple sclerosis, stroke, or musculoskeletal diseases, it can complement conventional therapeutic programs [17,18].

The primary objective of this randomized clinical trial was to evaluate the effects using the Nintendo Switch, with the video games Nintendo Switch Sports and Dr Kawashima’s Brain Training, combined with a conventional occupational therapy intervention to improve gross and fine motor skills, functionality, quality of life, and cognitive status, comparing the results to a control group in order to enhance healthy aging. In addition, the aim of this study was to assess the levels of satisfaction and compliance with the treatment.

## 2. Materials and Methods

### 2.1. Study Design

A randomized single-blind clinical trial is presented. All the participants came from the Physiocare Madrid centre and ORPEA nursing home in the Community of Madrid. The Consolidated Standards of Reporting Trials (CONSORT) 2010 checklist [19] was followed to ensure the methodological quality of this work.

This project was approved by the Research Ethics Committee of the Universidad Rey Juan Carlos (code: 2702202309423) prior to the start of the project. The ethical principles for medical research involving human subjects of the Declaration of Helsinki adopted at the 18th Assembly of the World Medical Association (WMA) (Helsinki, Finland, June 1964), along with the latest version revised at the 64th General Assembly of the WMA held in Fortaleza (Brazil) in October 2013, were followed. In addition, the study will comply with the legislations of our country, including Law 14/2007 on Biomedical Research and the Royal Decree 223/2004. All participants were informed of the objectives of this study and signed an informed consent form. Finally, the presented work was registered in Clinical Trials (NCT06129630).

### 2.2. Participants

All patients were recruited from the Physiocare Madrid rehabilitation centre and ORPEA nursing home in the community of Madrid. Patients had to meet the following inclusion criteria: subjects of both sexes; over 65 years of age; regularly attend rehabilitation treatment at the Physiocare Madrid centre or the ORPEA nursing home; do not present moderate or severe cognitive impairment (measured through the MEC-Lobo test with a score greater than 24 points); present an independence in basic activities of daily living (measured through the modified Barthel Index (Shah) with a score greater than 91 points); agree to voluntarily participate in the study and sign the informed consent.

The exclusion criteria for this study were as follows: any presence of neurodegenerative disease; any presence of physical illness that compromised the person’s safety; severe cognitive impairment. Both the inclusion and exclusion criteria were assessed by the centre’s geriatrician.

Participants who met these criteria were invited to participate in the study, and provided with the necessary information (benefits, risks, procedures, interventions, etc.) to make the decision to participate or not. Those who accepted then had to sign the informed consent form and were evaluated by occupational therapists externally to the study.

The randomization sequence was performed using the OxMaR program for the minimization and randomization of clinical studies [20]. The allocation sequence number of each subject was delivered in an opaque envelope and provided to the investigators.

Participants were randomly assigned to one of two intervention groups: (1) Experimental group (combination of conventional occupational therapy sessions and the Nintendo Switch); and (2) Control group (conventional occupational therapy sessions).

### 2.3. Intervention

After randomization and initial assessment, the intervention started immediately and lasted 8 weeks.

After being assigned to the intervention groups, all participants then attended occupational therapy sessions. Participants in the control group (CG) received conventional occupational therapy sessions two days a week for the maintenance of their cognitive and physical abilities, each lasting 60 min. Conventional occupational therapy included sessions of cognitive stimulation, psychomotor skills, reflective reading therapy, and upper limb functional activities, among others. The duration of the treatment was 8 weeks. Participants assigned to the experimental group (EG) received conventional occupational therapy sessions two days a week with a duration of 60 min (same as the CG), in addition to two sessions per week with the Nintendo Switch, lasting 50 min each. None of the EG members had previously used the Nintendo Switch.

The Nintendo Switch is a game console that was released in 2017 and was developed to be an improved version of the Nintendo Wii. The Nintendo Switch is more versatile, has better resolution, and is more compact. This console incorporates technological improvements such as the “Joy-cons” controllers that include an infrared camera which can capture the movement of the hands [18].

The EG’s Nintendo Switch sessions were conducted in a safe environment. The mode of use of the Nintendo Switch was using external controls or the “Joy-Con” wirelessly. Thanks to the integrated gyroscopes, accelerometers, and infrared motion sensors, these controllers allow the replication of the movement of a person. In addition, the image was projected onto a 54-inch digital whiteboard that allowed and facilitated the best visibility of the avatars and immersion in the video games.

For the Nintendo Switch sessions, Nintendo Switch Sports and Dr Kawashima’s Brain Training video games were used.

Nintendo Switch Sports:
Golf: requires the player to stand upright, perform block movements with both upper limbs and slight wrist movements, as well as requiring trunk and shoulder stability.Tennis: requires the player to perform upper limb movements, such as flexion-extension, abduction and adduction of the shoulder, flexion-extension of the elbow, and pronation-supination of the forearm, as well as testing strength and the player’s reflexes.Bowling: requires shoulder flexion-extension, elbow flexion-extension, and cylindrical palm pressing in order to press a button with the second finger until releasing it to throw.Chambara: consists of performing movements of flexion-extension, abduction and adduction of the shoulder, elbow flexion-extension, and the palmar pressure of the hand in order to defeat the opponent.


2.Dr Kawashima’s Brain Training for Nintendo Switch
Rock, paper, and scissors: the player must analyse what result they are asked for and perform the gesture of stone (closed fist), paper (open hand), or scissors (extension of the second and third fingers and the opposition of the first finger to the fourth and fifth fingers). With the other hand, the player needs to hold the other controller horizontally, so that the camera picks up the movement.Fingerprinting: consists of performing simple mental calculations and representing the result with the fingers of the hand (right or left), which can be one, two, three, four, five, or zero.Finger gymnastics: This consists of performing different hand gestures (right or left) in the shortest possible time. These gestures engage the opposition of the fingers to the palm and the extension of one or more fingers.Bird Counter: Two participants must count the birds that appear on the screen and enter the correct number before the opponent (by pressing the buttons with their thumb).Flags: Two participants must memorize the correct order of the raised flags and imitate it following the sequence shown. The movements involved in this game are flexion, extension, abduction, and shoulder adduction.Box counter: Two participants must memorize the number of boxes that appear on the screen before they disappear. Each player must enter the correct number of boxes (by pressing the buttons with their thumb) as quickly as possible before their opponent.


The Nintendo Switch routines were performed following the protocol specified in Figure 1.

### 2.4. Measures

The measurements began with an initial assessment where the following detailed assessment tests were administered. After finishing the intervention (after 8 weeks), a second evaluation was carried out to finalize the study.

General Medical Information

Data on the general health status of individuals will be extracted from the documentation provided by the participant; this will include sex, age, medical and social history, and the hours of weekly occupational therapy received.

2.Examination of Cognitive Status

Mini Examen Cognoscitivo-Lobo [21]: This assessment tool provides information regarding a person’s cognitive status. By means of the score obtained, the possible presence of cognitive impairment can be notified. In this test, the higher the score, the better cognitive state of the person.

3.Examination of Functional Independence

Barthel Index modified by Shah [22]: This scale is utilized to evaluate an individual’s degree of functional independence in fundamental activities of daily life. A higher score on this test indicates a greater level of functional independence for the individual. This test exhibits greater sensitivity and enhanced reliability when compared to the original Barthel Index.

4.Examination of the Upper Limb Function

Nine Hole Peg Test [23]: This test assesses fine manual dexterity and is widely regarded as a ‘gold standard’ in such evaluations [24]. It involves placing and removing a series of pegs on a board with nine holes as swiftly as possible. The test records the time taken by the individual for each upper limb. It detects progression over time, and is sensitive to changes induced by treatment. There is an excellent test–retest reliability of the right and left hand for subjects with and without impairment.

Box and Block Test [25]: This test evaluates the gross motor skills of the upper limbs. It involves a wooden box divided into two halves by an axis, with the objective of transferring as many wooden blocks as possible from one half to the other using each upper limb within a one-minute timeframe. The test records the number of blocks transferred by the individual with each upper limb within the allotted time. The Box and Block Test is simple and reliable in assessing gross manual dexterity, offering a useful measure to ensure objective upper limb functionality. This also ensure an excellent test–retest reliability of the right and left hand for older people with and within impairment.

5.Examination of quality of life

WHOQOL-BREF [26]: This questionnaire is designed for self-administration, and aims to evaluate overall perceptions of quality of life and health. It comprises the four following domains: physical health, psychological well-being, social relationships, and environmental factors. On this scale, a higher score indicates a higher quality of life. This also provides an excellent test–retest reliability of the right and left hand for older people.

6.Examination of satisfaction with the treatment received

The Client Satisfaction Questionnaire (CSQ-8) [27]: This self-administered questionnaire comprises eight questions aimed at evaluating the satisfaction level regarding the received care and service quality, along with the extent to which patient expectations prior to the intervention were met. Responses were coded on a scale of 1 to 4, yielding a total questionnaire score of 32 points. Higher scores reflect increased satisfaction with the treatment provided. In its adaptation to Spanish, it presents adequate psychometric properties and maintains the properties of the initial questionnaire. Therefore, it is suitable to assess satisfaction with the health service received in the Spanish population. The CSQ-8 will be applied to both treatment groups of this research [28].

### 2.5. Statiscal Analysis

Statistical analyses were performed using the SPSS 27.0 (Copyright© 2024 IBM SPSS Corp., Armonk, NY, USA). Non-normal distribution was confirmed by the Shapiro–Wilk test. Descriptive data were represented by mean and standard deviation (continuous variables) and frequency and percentage (categorical variables). Regarding the analysis using independent samples, the Kruskal–Wallis rank ANOVA test was used. Regarding the analysis of related samples, the Wilconxon test was used. For correlations between variables, we use the Spearmen Test. Additionally, the effect size of the differences was estimated by transforming Cohen’s d into a correlation coefficient (dr). Values of 0.20, 0.40, and 0.60 represent mild, moderate, and high effect sizes, respectively. The significant *p*-value was *p* < 0.05.

## 3. Results

### 3.1. Attrition Rate

The study concluded with 30 participants, with 15 being in each group. The flow diagram of the experiment is shown in Figure 2.

All the 30 enrolled participants successfully completed the eight-week intervention, resulting in a retention rate of 100%.

### 3.2. Baseline Characteristics of Participants

The descriptive data of the sample, as well as the tests administered are reported in Table 1. A total of 30 participants were divided into two groups, whose mean age was 88.83 ± 3.94, and 58.1% of the sample identified as female.

As indicated by the data, no significant differences were found at the baseline in the entire sample, except slightly in the age variable, with the experimental group being slightly larger than the control group (Table 1).

When performing the pre-post-intervention analysis according to the independent samples for the control group and the experimental group, significant differences were found in the vast majority of the variables analysed (Table 2), which highlights an improvement following the intervention, mainly in measures of skill and perceptions of quality of life. Similarly, the effect size shows values in line with the significant difference.

The analysis according to the related samples (Table 3), both in the control group and in the experimental group, shows that both groups obtained improvements following their respective interventions, with the experimental group benefiting the most, emphasised by the significant changes shown by all the variables.

To discover how the variables are related, a correlational analysis was carried out in which we highlighted a significant positive relationship between the Barthel and the BBT manual dexterity test and a significant negative relationship with the NHPT manual dexterity test (Table 4), as well as the negative relationship between age and the Barthel Index and a positive relationship with the CQS-8.

## 4. Discussion

To the best of our knowledge, this is the first single-blind randomized clinical trial to use the Nintendo Switch as a rehabilitation tool for older adults to promote healthy aging. The aim of the present study was to investigate the effects of the Nintendo Switch, with the combined use of Nintendo Switch Sports and Dr Kawashima’s Brain Training, and a conventional occupational therapy intervention in order to improve gross and fine motor skills, functionality, quality of life, and cognitive ability comparatively to a control group. Our results show that an eight-week period of using the Nintendo Switch results in improvements in gross and fine motor function, cognitive ability, functionality, and quality of life in the experimental group. High satisfaction and excellent attendance were achieved in both groups, being higher in the experimental group. In addition, the Nintendo Switch’s protocol of use did not generate adverse side effects, so this technology could be used to increase the level of satisfaction and compliance with traditional interventions in the elderly.

Currently, there are few published studies in which a VR intervention is performed with the Nintendo Switch being combined with conventional rehabilitation. In addition, there is no consensus on the duration of protocols with the use of the Nintendo Switch [18,29,30,31]. Takei et al. [29] developed a study to test the safety, feasibility, and acceptability of upper and lower limb strength training in older people with musculoskeletal conditions, using the Nintendo Switch video game Ring Fit Adventure combined with conventional physiotherapy for six sessions. The authors state that it is a safe and feasible tool as an adjunctive treatment to a rehabilitation program as long as it is supervised, at least during the early phases. In addition, the inclusion of new technologies leads to a greater enjoyment of the rehabilitation process. Blázquez-González et al. [30] carried out a protocol for the use of the Nintendo Switch alongside physiotherapy, occupational therapy, speech therapy, and neuropsychology sessions in stroke patients. Following a protocol of six VR sessions, the authors found benefits in the neurological and anxiety status of the participants, along with conventional rehabilitation. Kim et al. [31] combined the use of transcranial direct current stimulation and the use of the Nintendo Switch via the *1–2-Switch* video game during an eight-week protocol. The authors obtained improvements in fine and gross motor dexterity, grip strength, and upper limb functionality in stroke patients. For their part, Cuesta-Gómez et al. [18] developed an intervention project, combining the use of the Nintendo Switch and conventional occupational therapy sessions (sessions of mobilization, stretching and strengthening of the upper limb, dexterity, and functionality activities) in patients with multiple sclerosis. To complete this, Dr. Kawashima’s Brain Training video game was used for eight weeks, and improvements in grip strength, coordination, fine and gross motor function, executive functions, and upper limb functionality were observed, in addition to the protocol being valued by the participants with high satisfaction. The protocol that guides this work was carried out following the studies of Kim et al. [31] and Cuesta-Gómez et al. [18]. Our protocol was based on 16 sessions of 50 min, with a frequency of twice a week for eight weeks. The patients in this study report high satisfaction with the use of technology, as reflected in the results of the aforementioned publications. In addition, improvements in fine and gross motor skills, quality of life, cognitive status, and the functionality of older people have been observed in the present study, as in the previous studies. Participants received conventional occupational therapy combined with the Nintendo Switch protocol, meaning that the intragroup results should be interpreted as a combination of both types of therapies.

The Inclusion of VR in rehabilitation programs improves or avoids the worsening of some cognitive, physical, psychosocial, and quality of life abilities in older people [16,32,33]. Our results are in agreement with those of Chao et al. [34], Yen et al. [35], and García-Bravo et al. [16], where, through the use exergames with other commercial VR devices, such as the Nintendo Wii or the Kinect, positive influences are observed in the psychosocial field, in addition to improvements in depression, quality of life, postural control, and a positive impact on quality of life in older adults. This may be due to the playful component of exergames.

Promoting healthy aging in older people through rehabilitation programs can encourage improved habits and maintain the person’s independence [6,7]. Lifestyle alterations, as well as restrictions on participation in leisure activities caused by the loss of abilities in old age, can affect the psychological state or health and well-being of older people [36]. Therefore, our results show that, through the use of video games, older people can improve their perception of general, physical, and psychological health, as well as improve their motivation to participate in activities and the rehabilitation process, thus promoting healthy aging.

However, this study has several limitations. First, the participants are not from different parts of Spain, but they are at least from two different centres. We used a small sample size, meaning future studies should study the effects of our experimental protocol on a larger sample. In addition, participants could not have moderate or severe cognitive impairment due to the difficulty of using the Nintendo Switch. It would be interesting to conduct future studies with people with moderate cognitive impairment to promote healthy aging. Likewise, the results cannot be generalized to all older people (presence of other degenerative diseases, different degrees of cognitive impairment, different levels of education). Finally, not all participants were institutionalized in a residence as some of them lived in the community. Therefore, future studies should study unifying the criteria for choosing the sample with respect to their place of residence.

## 5. Conclusions

Our results show that an eight-week experimental protocol, using Dr. Kawashima’s Brain Training and Nintendo Switch Sports for the Nintendo Switch, combined with a conventional occupational therapy intervention, showed improvements in the fine and gross motor dexterity, cognitive and functional status, and perceived quality of life in older people. We want to emphasize the importance of these results in the context of promoting healthy aging and the treatment of cognitive impairment. Therefore, the use of the Nintendo Switch as a complement to occupational therapy can be a good option to promote healthy aging in the elderly.

## Figures and Tables

**Figure 1 healthcare-12-00672-f001:**
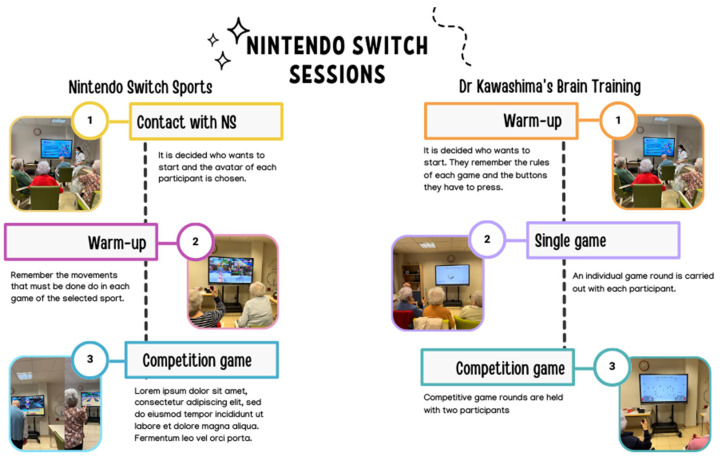
Nintendo Switch session protocol.

**Figure 2 healthcare-12-00672-f002:**
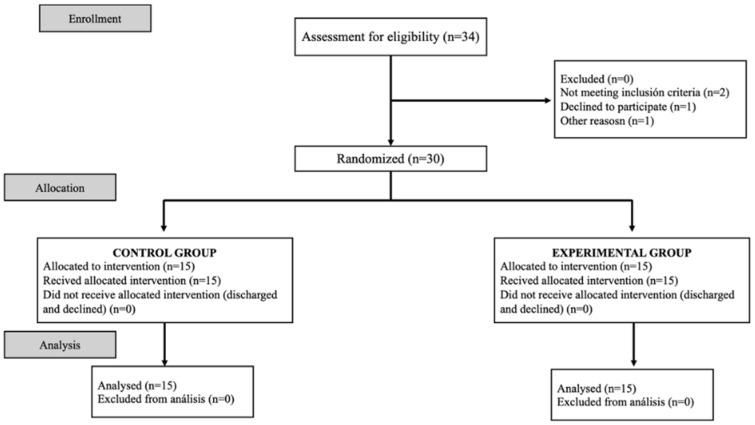
Flow diagram of the experimental procedure.

**Table 1 healthcare-12-00672-t001:** Descriptive data of the sample (*n* = 30) at baseline before the intervention.

	Full Sample	Control Group (*n* = 15)	Experimental Group (*n* = 15)	H	Sig. *p*
Gender (Fr (%))					
Female	18 (58.1)	9 (60)	9 (60)		
Male	12 (38.7)	6 (40)	6 (40)		
Age (Mean ± DS)	88.83 ± 3.94	87.47 ± 4.8	90.2 ± 2.27	3.76	0.052
Barthel Index (Mean ± DS)	95.00 ± 3.09	95.40 ± 3.43	94.6 ± 2.77	0.449	0.503
MEC-Lobo (Mean ± DS)	29.73 ± 2.72	29.93 ± 2.81	29.53 ± 2.72	0.158	0.691
NHPT Right (Mean ± DS)	39.85 ± 15.7	44.38 ± 18.19	35.32 ± 11.66	1.81	0.178
NHPT Left (Mean ± DS)	43.89 ±19.81	49.87 ± 25.32	37.91 ± 9.77	1.93	0.165
BBT Right (Mean ± DS)	41.23 ± 10.8	40 ± 11.408	42.47 ± 10.4	0.414	0.520
BBT Left (Mean ± DS)	39.90 ± 9.901	39.2 ± 10.7	40.6 ± 9.32	0.209	0.647
General Perception Quality of Life (Mean ± DS)	2.67 ± 0.802	2.67 ± 0.72	2.67 ± 0.9	0.000	0.982
General Perception Health (Mean ± DS)	2.17 ± 0.648	2.07 ± 0.704	2.27 ± 0.594	0.630	0.427
Physical Health (Mean ± DS)	2.17 ± 0.648	2.07 ± 0.7	2.27 ± 0.59	0.021	0.884
Psychological Health (Mean ± DS)	12.85 ± 2.36	12.76 ± 2.65	12.95 ± 2.12	0.011	0.915
Social Relations (Mean ± DS)	12.77 ± 2	12.73 ± 2.13	12.82 ± 1.92	0.610	0.435
Environment (Mean ± DS)	12.44 ± 2.04	12.16 ± 2.12	12.72 ± 1.98	0.074	0.785
CSQ-8	25.43 ± 4.11	22 ± 1.73	28.87 ± 2.61	18.91	0.000

H = Test value, Kruskal–Wallis rank ANOVA.

**Table 2 healthcare-12-00672-t002:** Post-intervention independent sample analysis.

	Control Group (*n* = 15) (Mean ± DS)	Experimental Group (*n* = 15) (Mean ± DS)	H	Sig. *p*	d_r_
Barthel Index	29.33 ± 3.01	30.27 ± 3.01	0.506	0.477	0.31
MEC-Lobo	43.94 ± 18.9	33.67 ± 11.44	0.739	0.39	0.65
NHPT Right	51.67 ± 30.17	36.9 ± 9.19	3.10	0.078	0.66
NHPT Left	38.87 ± 10.65	50.07 ± 12.34	2.29	0.130	0.97
BBT Right	38.4 ± 12.35	48 ± 11.47	4.58	0.032	0.80
BBT Left	2.2 ± 0.86	3.47 ± 0.64	3.97	0.046	1.67
General Perception Quality of Life	2.07 ± 0.59	3 ± 0.65	12.61	0.000	1.49
General Perception Health	12.07 ± 2.61	14.3 ± 2.16	11.32	0.001	0.94
Physical Health	11.91 ± 1.99	14.61 ± 2.07	5.57	0.018	1.32
Psychological Health	11.32 ± 1.95	14.06 ± 2.15	9.83	0.002	1.33
Social Relations	12.06 ± 1.82	15.03 ± 2.15	9.87	0.002	1.49
Environment	29.33 ± 3.01	30.27 ± 3.01	10.69	0.001	0.31

H = Test value, Kruskal–Wallis rank ANOVA.

**Table 3 healthcare-12-00672-t003:** Analysis of the effect of the intervention on related samples.

	Control Group					Experimental Group				
	Pre (Mean ± DS)	Post (Mean ± DS)	Z	Sig.	d_r_	Pre (Mean ± DS)	Post (Mean ± DS)	Z	Sig.	d_r_
Barthel Index	95.40 ± 3.43	29.33 ± 3.01	−2.442	0.015	0.99	94.6 ± 2.77	30.27 ± 3.01	−2.814	0.005	0.99
MEC-Lobo	29.93 ± 2.81	43.94 ± 18.9	−2.714	0.007	0.46	29.53 ± 2.72	33.67 ± 11.44	−1.935	0.053	0.24
NHPT Right	44.38 ± 18.19	51.67 ± 30.17	−0.682	0.496	0.14	35.32 ± 11.66	36.9 ± 9.19	−2.442	0.015	0.07
NHPT Left	49.87 ± 25.32	38.87 ± 10.65	−1.079	0.281	0.27	37.91 ± 9.77	50.07 ± 12.34	−1.988	0.047	0.47
BBT Right	40 ± 11.40	38.4 ± 12.35	−0.915	0.360	0.06	42.47 ± 10.4	48 ± 11.47	−2.955	0.003	0.24
BBT Left	39.2 ± 10.7	2.2 ± 0.86	−0.697	0.486	0.92	40.6 ± 9.32	3.47 ± 0.64	−3.270	0.001	0.94
General Perception Quality of Life	2.67 ± 0.72	2.07 ± 0.59	−2.121	0.034	0.41	2.67 ± 0.9	3 ± 0.65	−2.762	0.006	0.20
General Perception Health	2.07 ± 0.70	12.07 ± 2.61	0.000	1.000	0.93	2.27 ± 0.59	14.3 ± 2.16	−3.051	0.002	0.96
Physical Health	2.07 ± 0.70	11.91 ± 1.99	−2.585	0.010	0.95	2.27 ± 0.59	14.61 ± 2.07	−2.764	0.006	0.97
Psychological Health	12.76 ± 2.65	11.32 ± 1.95	−2.294	0.022	0.29	12.95 ± 2.12	14.06 ± 2.15	−2.728	0.006	0.25
Social Relations	12.73 ± 2.13	12.06 ± 1.82	−2.392	0.017	0.16	12.82 ± 1.92	15.03 ± 2.15	−2.451	0.014	0.47
Environment	12.16 ± 2.12	29.33 ± 3.01	−1.847	0.065	0.95	12.72 ± 1.98	30.27 ± 3.01	−2.963	0.003	0.96

Z = Test value, Wilconxon test.

**Table 4 healthcare-12-00672-t004:** Correlations between variables.

	Age	Gender	BI	MEC-Lobo	NHPT-R	NHPT-L	BBT-R	BBT-L	GPQoL	GPH	PH	PsH	SR	E	CSQ-8
Age	0.178	−0.095	−0.374 *	−0.151	−0.055	0.202	0.064	−0.062	−0.053	0.171	0.024	−0.082	0.104	0.416	0.178 *
Gender	0.178	1.000	0.143	−0.218	−0.039	0.008	−0.051	0.028	−0.008	0.009	0.122	0.165	0.012	0.211	0.104
BI	−0.095	0.143	1.000	0.260	−0.370 *	−0.442 *	0.385 *	0.547 **	0.175	0.041	0.030	0.245	0.175	0.256	−0.007
MEC-Lobo	−0.374 *	−0.218	0.260	1.000	0.069	0.019	−0.057	0.041	0.235	0.090	0.071	−0.014	0.087	−0.091	0.095
NHPT-R	−0.151	−0.039	−0.370 *	0.069	1.000	0.878 **	−0.827 **	−0.780 **	−0.122	−0.259	0.190	0.141	0.077	0.072	−0.259
NHPT-L	−0.055	0.008	−0.442 *	0.019	0.878 **	1.000	−0.782 **	−0.785 **	−0.121	−0.211	0.251	0.189	0.211	0.160	−0.288
BBT-R	0.202	−0.051	0.385	−0.057	−0.827 **	−0.782 **	1.000	0.943 **	0.142	0.236	−0.068	−0.185	−0.084	−0.060	0.246
BBT-L	0.064	0.028	0.547 **	0.041	−0.780 **	−0.785 **	0.943 **	1.000	0.127	0.296	−0.052	−0.107	−0.073	−0.007	0.234
GPQoL	−0.062	−0.008	0.175	0.235	−0.122	−0.121	0.142	0.127	1.000	0.397 *	0.349	0.106	0.442 *	0.455 *	−0.084
GPH	−0.053	0.009	0.041	0.090	−0.259	−0.211	0.236	0.296	0.397 *	1.000	0.090	0.060	0.246	0.285	0.011
PH	0.171	0.122	0.030	0.071	0.190	0.251	−0.068	−0.052	0.349	0.090	1.000	0.317	0.355	0.580 **	−0.065
PsH	0.024	0.165	0.245	−0.014	0.141	0.189	−0.185	−0.107	0.106	0.060	0.317	1.000	0.668 **	0.772 **	−0.124
SR	−0.082	0.012	0.175	0.087	0.077	0.211	−0.084	−0.073	0.442 *	0.246	0.355	0.668 **	1.000	0.819 **	−0.039
E	0.104	0.211	0.256	−0.091	0.072	0.160	−0.060	−0.007	0.455 *	0.285	0.580 **	0.772 **	0.819 **	1.000	−0.134
CSQ-8	0.416 *	0.104	−0.007	0.095	−0.259	−0.288	0.246	0.234	−0.084	0.011	−0.065	−0.124	−0.039	−0.134	1.000

BI = Barthel Index; NHPT-R = NHPT Right Hand; NHPT-L = NHPT Left Hand; BBT-R = BBT Right Hand; BBT-L = BBT Left Hand; GPQoL = General Perception Quality of Life; GPH = General Perception Health; PH = Physical Health; PsH = Psychological Health; SR = Social Relationships; E = environment. * = *p* < 0.05, ** = *p* < 0.01.

## Data Availability

The data presented in this study are available on request from the corresponding author.

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
