# Peer review of "The Effectiveness and Usability of a Rehabilitation Program Using the Nintendo Switch to Promote Healthy Aging in Older People with Cognitive Impairment: A Randomized Clinical Trial"

_healthcare, 2024, doi:10.3390/healthcare12060672_

Round 1
Reviewer 1 Report
Comments and Suggestions for Authors
The submitted manuscript titled "Effectiveness and Usability of a Rehabilitation Program through the Nintendo Switch to promote Healthy Aging in Older People with Cognitive Impairment: A Randomized Clinical Trial" presents the results of a study aimed at evaluating the effectiveness of a rehabilitation program using the Nintendo Switch in promoting healthy aging among older individuals with cognitive impairment. The study utilized a randomized clinical trial design and included measures of motor skills, cognitive status, functionality, quality of life, and satisfaction with the intervention. Results indicate significant improvements in various outcome measures following an eight-week intervention period, suggesting that the use of the Nintendo Switch, combined with conventional occupational therapy, can positively impact the well-being of older adults with cognitive impairment.
Here are some critiques and suggestions for revision to increase the impact of the paper:
- Clarity of Introduction: While the introduction provides a comprehensive overview of healthy aging and the rationale for the study, it could benefit from more concise wording and clearer organization. Try to streamline the introduction to highlight the gap in research and the specific objectives of the study.
- Methodological Details: The manuscript provides detailed information on the study design, participant recruitment, intervention protocols, and outcome measures. However, it would be helpful to include more information on the randomization process, such as how randomization was concealed and any measures taken to minimize bias.
- Participant Characteristics: The baseline characteristics of participants are well-described, but it's essential to ensure that the sample is representative of the target population. Consider discussing any demographic factors that may have influenced the results, such as age distribution or comorbidities.
- Outcome Measures: The manuscript includes a variety of outcome measures to assess the effectiveness of the intervention. However, it would be beneficial to discuss the clinical relevance of these measures and how improvements in these outcomes translate to meaningful changes in participants' daily lives.
- Figures: The figures included in the manuscript effectively illustrate key concepts and results, enhancing the clarity and understanding of the findings. They provide visual representations of data, facilitating interpretation for readers.
- Tables: The tables are well-organized and clearly present essential information, such as participant characteristics, baseline measures, and post-intervention outcomes. They contribute to the overall readability of the manuscript and assist readers in navigating complex data.
- Statistical Analysis: The statistical analysis section provides information on the tests used and the significance criteria. Ensure that the statistical methods are appropriate for the study design and outcomes measured. Additionally, consider including effect sizes or confidence intervals to provide a more comprehensive understanding of the results.
- Discussion: The discussion effectively summarizes the findings and compares them to previous research. However, it could be strengthened by discussing potential mechanisms underlying the observed effects and addressing limitations in more detail. Additionally, consider implications for clinical practice and future research directions.
- Conclusions: The conclusions section succinctly summarizes the key findings of the study. Consider reiterating the importance of the research findings in the context of promoting healthy aging and addressing cognitive impairment.
As for conclusion,
This is a promising study to give some valuable information about the cognitive rehabilitation of the older population to postpone potential process of the dementia and increase quality of life reducing promising comorbidities. After mentioning issue revision it could be acceptable for the publication.
Author Response
We would like to thank the editor and reviewers for their comments in this review, which have greatly improved the readability of the manuscript. We would like to inform you that we have edited the manuscript according to the very constructive suggestions from the reviewers. We have highlighted (yellow) all the changes we have made throughout the text.
Below, please find a list of revisions and a response to each of the reviewer’s comments. We hope that the revisions in the manuscript and our accompanying responses will be enough to make our manuscript suitable for publication in the HEALTHCARE.
We shall look forward to hearing from you at your earliest convenience.
Yours sincerely,
The authors

Reviewer 2 Report
Comments and Suggestions for Authors
Dear authors,
The manuscript is very interesting and the statements and objectives are clear. However, the statistical section is a huge concern. As you might know, the multiple testing is an important source of bias in RCTs. Thus, I strongly recommend you use the factorial ANOVA (see the attached document) to perform your analysis. Accordingly, the Cohen d effect size would be welcome to ensure clinical value of your comparisons. Those issues may change the outcomes and the observed results.
Aside from that, the manuscript is excellent. Congrats.

Author Response

(The authors gave the same response as above.)

Reviewer 3 Report
Comments and Suggestions for Authors
This study touches an important issue in the today life. People are getting old and older and many health/nursing care systems are facing nursing shortage and the aging in place paradigm is getting more importance. Though, elderly living in nursing houses are facing also many issues and challenges. Care staffs are overbooked, etc.
The quality of elderly people living in nursing houses was discussed in several passt studies and thus shows the importance of the topic in the field of healthy aging in the institutional nursing houses.
This study investigates the effects of using the Nintendo Switch on the quality of aging amongst others.
I appreciate the research design.
The paper needs, althrough its strengh, some improvement
1. figure (1 & 2) resolution: to be improved
2. revise table 3 and 4 description. For what stand **?
3. A short description of Nintendo Switch system will help the reader a bit more. The paper only lists the feature without describing the system. It is valid for Dr Kawashima's Brain Training
Author Response

(The authors gave the same response as above.)

Round 2
Reviewer 2 Report
Comments and Suggestions for Authors
Dear authors,
I still insist that the multiple comparison issue is important. Please, consider another statistical method to compare the results.
Cheers
Author Response
We would like to thank the editor and the reviewer for give us another opportunity. We have highlighted (yellow) all the changes we have made throughout the text.
We shall look forward to hearing from you at your earliest convenience.
Yours sincerely,
The authors
